# Estimating the trend of COVID-19 in Norway by combining multiple surveillance indicators

**Gunnar Rø**[1]*, **Trude Marie Lyngstad**[1], **Elina Seppälä**[1], **Siri Nærland Skodvin**[2], **Lill Trogstad**[1], **Richard Aubrey White**[1], **Arve Paulsen**[3], **Trine Hessevik Paulsen**[1], **Trine Skogset Ofitserova**[1], **Petter Langlete**[1], **Elisabeth Henie Madslien**[1], **Karin Nygård**[1], **Birgitte Freisleben de Blasio**[1,4]

**1** Norwegian Institute of Public Health, Division of Infection Control, Oslo, Norway, **2** Centre for Fertility and Health, Norwegian Institute of Public Health, Oslo, Norway, **3** Norwegian Directorate of Health, Oslo, Norway, **4** Oslo Centre for Biostatistics and Epidemiology, University of Oslo, Oslo, Norway

* gunnaroyvindisaksson.ro@fhi.no

**Data Availability Statement:** All data and code can be found at: https://github.com/folkehelseinstituttet/covid19_trend.

## Abstract

Estimating the trend of new infections was crucial for monitoring risk and for evaluating strategies and interventions during the COVID-19 pandemic. The pandemic revealed the utility of new data sources and highlighted challenges in interpreting surveillance indicators when changes in disease severity, testing practices or reporting occur. Our study aims to estimate the underlying trend in new COVID-19 infections by combining estimates of growth rates from all available surveillance indicators in Norway. We estimated growth rates by using a negative binomial regression method and aligned the growth rates in time to hospital admissions by maximising correlations. Using a meta-analysis framework, we calculated overall growth rates and reproduction numbers including assessments of the heterogeneity between indicators. We find that the estimated growth rates reached a maximum of 25% per day in March 2020, but afterwards they were between -10% and 10% per day. The correlations between the growth rates estimated from different indicators were between 0.5 and 1.0. Growth rates from indicators based on wastewater, panel and cohort data can give up to 14 days earlier signals of trends compared to hospital admissions, while indicators based on positive lab tests can give signals up to 7 days earlier. Combining estimates of growth rates from multiple surveillance indicators provides a useful description of the COVID-19 pandemic in Norway. This is a powerful technique for a holistic understanding of the trends of new COVID-19 infections and the technique can easily be adapted to new data sources and situations.

## Introduction

For risk assessment and management of the COVID-19 pandemic, one of the main surveillance objectives was to produce incidence trends. Key indicators used were incidence of COVID-19 cases and hospital admissions In Norway, both formal [1–3] estimates of growth rates and effective reproduction numbers [4, 5] as well as qualitative assessments of

**Funding:** The author(s) received no specific funding for this work.

**Competing interests:** The authors have declared that no competing interests exist.

surveillance data were used to describe trends [3]. The quality of the data sources changed over time due to a large range of factors, for example during early parts of 2022 in Norway there were large changes in the testing regime and the risk of hospitalisation given infection decreased during the unfolding Omicron wave [6]. Therefore, a new and broader surveillance approach based on a larger range of surveillance indicators was implemented.

During the COVID-19 pandemic, trends have been formally estimated using both reproductive numbers [7, 8], *R*, and exponential growth rates [9], *r*. Reproduction numbers are defined as the average number of secondary cases infected by each infected, while by the exponential growth rate we refer to the factor, *r*, multiplying time, *t* in the exponent of the growth of cases over time $I(t) = I_0 exp(rt)$. While these quantities are related through the generation time [10], there are benefits and drawbacks of focusing on one or the other [11]. Growth rates provide a better understanding of the short-term development of an indicator, while reproduction numbers provide information about long-term evolution and the effect of potential interventions. However, reproduction numbers are model-dependent and influenced by several factors, with the most important factor being the shape of the generation time distribution, but also other factors like the number of cases infected abroad will impact estimates. Overall, both measures of trend are useful and in Norway, we estimated both quantities. In this paper we focus on growth rates as we want to compare the short-term trend of multiple different indicators.

In order to track the trend of the pandemic many new data sources have been used internationally. Wastewater surveillance [12] has been the most widely used, along with data on self-reported symptoms and test results collected though pre-existing cohorts or self-selected participation using smartphone apps have provided valuable information [13–15]. Repeated point prevalence studies have provided the gold standard estimates of prevalence and trends in new infections [16, 17]. Syndromic surveillance systems, for example based on visits to emergency rooms, and mortality have also been used to understand the spread of COVID-19. When interpreting these new indicators, it is important to understand how they relate to each other and to the underlying disease epidemiology both in terms of correlation and delays [18]. Multiple factors of credibility, timeliness, coverage and relation to new infections need to be evaluated.

In this paper, we present an analysis of combining growth rate estimates from multiple surveillance indicators during the COVID-19 pandemic in Norway. This approach introduces a novel composite growth rate that integrates information from various data sources with minimal modeling assumptions. The composite indicator mitigates the impact of variability in the relationship between individual indicators and the underlying disease dynamics, as well as fluctuations in data quality. This provides a more comprehensive and robust depiction of disease transmission, especially in complex and evolving epidemic contexts. Our aim is to outline the framework for data integration and offer a resilient method for analyzing disease dynamics with minimal assumptions and without ranking individual indicators based on their relative importance or historical performance. The approach is an extended, retrospective version of an analysis that has been performed in real-time as part of routine surveillance of COVID-19 in Norway since February 2022 [3].

## Materials and methods

### Data sources

We included ten key surveillance indicators used for surveillance during the COVID-19 pandemic as detailed in Table 1. These indicators can be classified in two main categories, describing either an incidence or a prevalence. The incidence data type refers to data sources that

**Table 1. Overview of the surveillance indicators used to estimate growth rates in Norway between March 2020 and the end of 2023.**

| Short Name | Indicator | Available | Reporting Frequency | Date | Type | Source |
|---|---|---|---|---|---|---|
| Wastewater | Relative concentration of SARS-CoV-2 RNA in the wastewater | May 2022—Oct 2023 | Weekly | Sampling week | Prevalence | Wastewater Surveillance Project at Norwegian Institute of Public Health (NIPH) |
| Proportion Positive—Survey | Proportion of symptomatic participants reporting positive tests for SARS-CoV-2 | Nov 2020—Dec 2023 | Weekly | Sampling week | Prevalence | Participatory surveillance system named "Symptometer" |
| Proportion Positive—Cohort | Proportion of respondents reporting positive SARS-CoV-2 tests | Jan 2022—June 2022 | Daily | Estimated symptom onset date | Incidence | Norwegian Mother, Father and Child Cohort Study(MoBa) |
| GP Consultations | Proportion of general practitioner (GP) consultations due to confirmed or suspected COVID-19 | Feb 2020—Dec 2023 | Weekly | Week of Consultation | Prevalence | Norwegian Syndromic Surveillance System(NorSySS) |
| Positive RAT-tests | Number of self-reported SARS-CoV-2 positive rapid antigen tests(RAT) | Jan 2022—Mar 2022 | Daily | Test Date | Incidence | Norwegian Directorate of Health |
| Hospital Prevalence | Proportion of all acute hospitalisations with, but not due to COVID-19 | March 2020—Sep 2023 | Daily | Admission Date | Prevalence | Norwegian Intensive and Pandemic Registry(NoPaR) and Norwegian Patient Registry(NPR) |
| Cases | Number of laboratory confirmed COVID-19 cases | Feb 2020—Dec 2023 | Daily | Test date | Incidence | Norwegian Surveillance System for Communicable Diseases (MSIS) |
| Proportion Positive | New l Laboratory confirmed COVID-19 cases divided by the number of test events | Feb 2020—Dec 2023 | Daily | Test date | Incidence | Norwegian Surveillance System for Communicable Diseases (MSIS and the MSIS laboratory database) |
| Hospital Admissions | Number of admissions to hospital with COVID-19 as the main cause | March 2020—Sep 2023 | Daily | Admission date | Incidence | Norwegian Intensive and Pandemic Registry(NoPaR) |
| Deaths | Number of COVID-19 associated deaths | Feb 2020-Dec 2023 | Weekly | Week of death | Incidence | Norwegian Cause of Death Registry |

record new events related to a SARS-CoV-2 infection, such as a positive test results or a hospital admissions. Prevalence data types are data that measures the current number or proportion of people who are infected or would test positive.

**Registry data.** We used data from several national health registries in Norway, many of which were available through the Emergency preparedness registry for COVID-19 (Beredt C19) [19]. Reports of laboratory confirmed COVID-19 cases were retrieved from the Norwegian Surveillance System of Communicable Diseases (MSIS and the MSIS laboratory database) [20, 21], only the first positive test per person in a given time period was include to avoid including multiple retests. This time period changed somewhat during the pandemic, but was always at least 60 days. For the Proportion of positive tests we use the number of positive cases as the numerator and the number of test-events, defined as one or more tests for SARS-CoV-2 PCR test or antigen test within 7 days as the denominator. Hospitalisations with COVID-19 as the main cause were retrieved from the Norwegian Intensive Care and Pandemic Registry [22] where the main cause of admission was determined by the clinician. The number of COVID-19 associated deaths were extracted from the Cause of Death Registry [23] with COVID-19 as the underlying or contributing cause of death (International Classification of Diseases, 10th edition (ICD-10) codes: U01.1, U07.2, U09.9 and U10.9). The hospital prevalence indicator is based on the proportion of acute admissions registered in the Norwegian Patient Register [24] who had a COVID-19 diagnosis code, but who were not admitted with COVID-19 as the main cause of the admission. During the pandemic most acutely admitted patients were tested for COVID-19, so this indicator gives an estimate of

the COVID-19 prevalence among the population who are admitted to hospital for non-COVID-19 causes.

The Norwegian syndromic surveillance system (NorSySS) [25] is based on consultations in primary health care from the Norwegian Registry for Primary Health Care [24]. Every consultation is coded using an International Classification of Primary Care, 2nd edition (ICPC-2) code. We use the number of consultations with codes R991 ("Suspected COVID-19"), R992 ("Confirmed COVID-19") and R33 ("COVID-19 Test") compared to the total number of consultations as a prevalence measure of COVID-19.

**Self-reported SARS-CoV-2 positive rapid antigen tests.** From late January 2022 to March 2022, the Norwegian population was asked to register positive self-administered rapid antigen tests(RAT) to their municipality of residence digitally. This data was then reported to the Norwegian Directorate of Health. We used the number of positive tests by testing date as an incidence measure.

**Wastewater.** From June 2022 to November 2023 semiweekly wastewater samples were tested for SARS-CoV-2 [26]. Samples were taken from municipal wastewater treatment plants in the largest cities in Norway, initially with 12 different sites(covering 30% of the population) and then with 5 sites from December 2022(coverage 25%). From April 2023 the number of sites were further downscaled to include 3 sites (coverage 22%). We used the fraction of detected SARS-CoV-2 RNA to the amount of the faecal indicator Pepper Mild Mottle Virus RNA (PMMoV) as a crude measure of COVID-19 prevalence.

**Participatory surveillance.** "Symptometer" was a participatory surveillance system established in November 2020 with weekly questionnaires to monitor symptoms and testing behaviour during the pandemic [27]. Starting up, the panel included a representative sample of the Norwegian population consisting of roughly 36 000 individuals, of which approximately 19 000 responded weekly (50%). The number of participants and response rates gradually decreased, and at the end of the study period there were approximately 22 000 participants with an average response rate of 19% (4200). In addition to symptom reporting for the preceding seven days, information about testing for SARS-CoV-2 was also reported, including test result and whether tested with self test/RAT or through health care services.

**Norwegian Mother, Father and Child Cohort Study.** The Norwegian Mother, Father, and Child Cohort Study (MoBa) is a pregnancy-based cohort that recruited participants between 1999 and 2008 [28]. The participation rate among pregnant women was 41%, and the cohort includes approximately 95,000 mothers, 75,000 fathers, and 114,000 children. Since March 2020, active adult participants have been invited to answer electronic questionnaires with questions related to the COVID-19 pandemic every 14–30 days, covering topics such as the onset of symptoms and testing activity [29]. The number of responses was high throughout the pandemic, with between 100 000 and 70 000 respondents.

From January 2022 to June 2022 information about the incidence of self-reported positive SARS-CoV-2 tests in the MoBa cohort was systematically extracted from the questionnaires. If a participant reported a positive test result along with a test date, we used the reported date directly. If no test date was provided, we estimated it based on the reported onset of symptoms. If neither a test date nor symptoms were reported, a test date was randomly sampled from the period covered by the questionnaire.

## Estimating growth rate

Our goal is to estimate the exponential growth rate over time, $r(t)$. With a time-series of incidence measurements, $I(t)$, we estimate the growth rate by the logarithmic derivative of a

suitably smoothed version of this time series, $S[I(t)]$

$$r(t) = \frac{dlog(S[I(t)])}{dt}$$

We smooth the incidence by the best fitting exponential function over a rolling window of length $2l + 1$ centred at $t$ by using a negative binomial regression with dispersion $\kappa$ on the incidence data in the time-window $[t - l, t + l]$.

$$I(t) \sim NegBin(exp(b_0 + r(t) * \tau + b_1 * \text{weekend}), \kappa), \tau \in [t - l, t + l]$$

where $b_0$, $r(t)$ and $b_1$ are regression coefficients and we include a possible weekend effect. The amount of smoothing is controlled by the length of the time-window, $2l + 1$. In the main results we use a window length of 25 days and in the S1 File we include sensitivity analysis for other choices of $l$.

For indicators providing daily incidence data, we use this regression model directly. For deaths, where there are only aggregated weekly data, we first disaggregate the data to a daily frequency using a Bayesian Gaussian Process model, similar to the model for estimating incidence from prevalence, which will be discussed in the following. For the "Proportion Positive" and the "Proportion Positive—Cohort" indicators we include the total number of responses or test events as an offset in the negative binomial model.

In the case of indicators measuring prevalence we first estimate incidence curves from the prevalence data and then use the regression model to calculate the growth rates. To ensure that we propagate the uncertainty we generate multiple sampled incidence trajectories and estimate the growth rates for each trajectory. We combine all the estimates from the different incidence trajectories into an overall growth rate by sampling from the sampling distribution of the regression coefficients. We use two methods for estimating the prevalence. For the "GP Consultations" and "Hospital Prevalence" indicators", the prevalence is given as a simple proportion of COVID-19 consultations or admissions compared to all consultations or admissions. For the "Proportion Positive—Survey" indicator we want to estimate the symptomatic prevalence of those surveyed, however, not everyone with symptoms chose to get tested. We therefore assume that the test positivity rate is equal among all participants who report having cold symptoms and use a Bayesian model for estimation.

Once we have estimated the prevalence, we estimate the incidence following [30]. Similarly following [31], we estimate relative incidence from wastewater data using the same approach as when estimating incidence from prevalence. See the S1 File for more details.

With available estimates of the growth rates over time for each indicator we want to evaluate potential systematic delays between them. We use hospital admissions as a reference and estimate a relative delay for the other indicators by finding the constant time-shift that maximises the Pearson correlation with the growth rate based on hospitalisations. Hospital admissions was chosen as the reference since it was one of the key indicator used to track the pandemic in real time. We find the optimal time-shift for each indicator and year separately and globally for all four years. For the rest of the analysis, we use the growth rates obtained after shifting by the estimated yearly delays. Next, we estimate the Pearson correlation matrix between all the data sources taking into account the uncertainty by using a Monte-Carlo approach where we sample from the sampling distribution of the growth rates and calculate the correlation coefficient with uncertainty in the overlapping time periods.

We combine samples from the individual growth rates from each indicator and estimate an overall growth rate using a meta-analytic framework with random effects and smoothing over time. With this approach we can also estimate the heterogeneity, $I^2$, quantifying how much of

the uncertainty is captured by the sampling uncertainty of the individual growth rates. A high $I^2$ value indicates that there is significant heterogeneity between the data sources that is not just due to the sampling uncertainty. The combined growth rate can then be translated to a reproduction number [10] by considering how the generation time has changed during the pandemic. We also calculate an associated relative incidence based on the combined growth rate.

More details are presented in the S1 File. The procedures described above were implemented using the R-programming language [32], with all the Bayesian methods implemented using the RStan package [33]. All the code and data are available at http://github.com/folkehelseinstitutet/covid19_trend. The study only used aggregated, de-identified and published surveillance data. For this reason, it did not require ethics committee approval.

## Results

In Fig 1 we show the weekly aggregated indicators on a logarithmic scale for all the included surveillance indicators. The figure shows the multiple waves of the COVID-19 epidemic in Norway and highlights which time periods the different indicators were available. Fig 2 shows the estimated time shifts for each indicator that maximises the correlation with hospitalisations. Most of the indicators have a positive shift indicating that they occur prior to hospitalisations, while deaths have a negative shift indicating that they are delayed compared with hospitalisations. There is variation in the time-shifts by year, in some cases the variation is significant, for example for the GP Consultations and the indicators for cases and proportion positive cases. The figure also shows that the correlations between the various indicators and hospitalisations can vary from year to year.

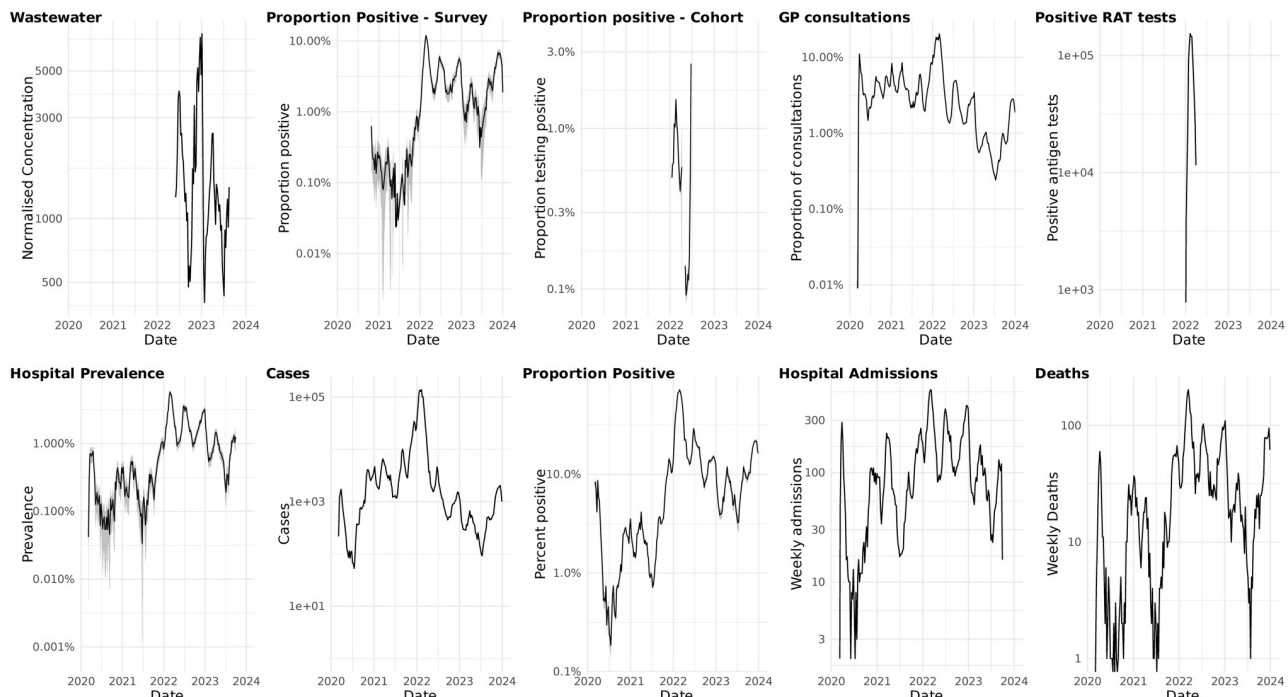

**Fig 1. Weekly aggregated data for all the included surveillance indicators plotted on a logarithmic scale.** For the prevalence-type indicators we include a 95% confidence interval for the estimated proportions, Norway 2020–2023. RAT: rapid antigen tests, GP: general practitioner.

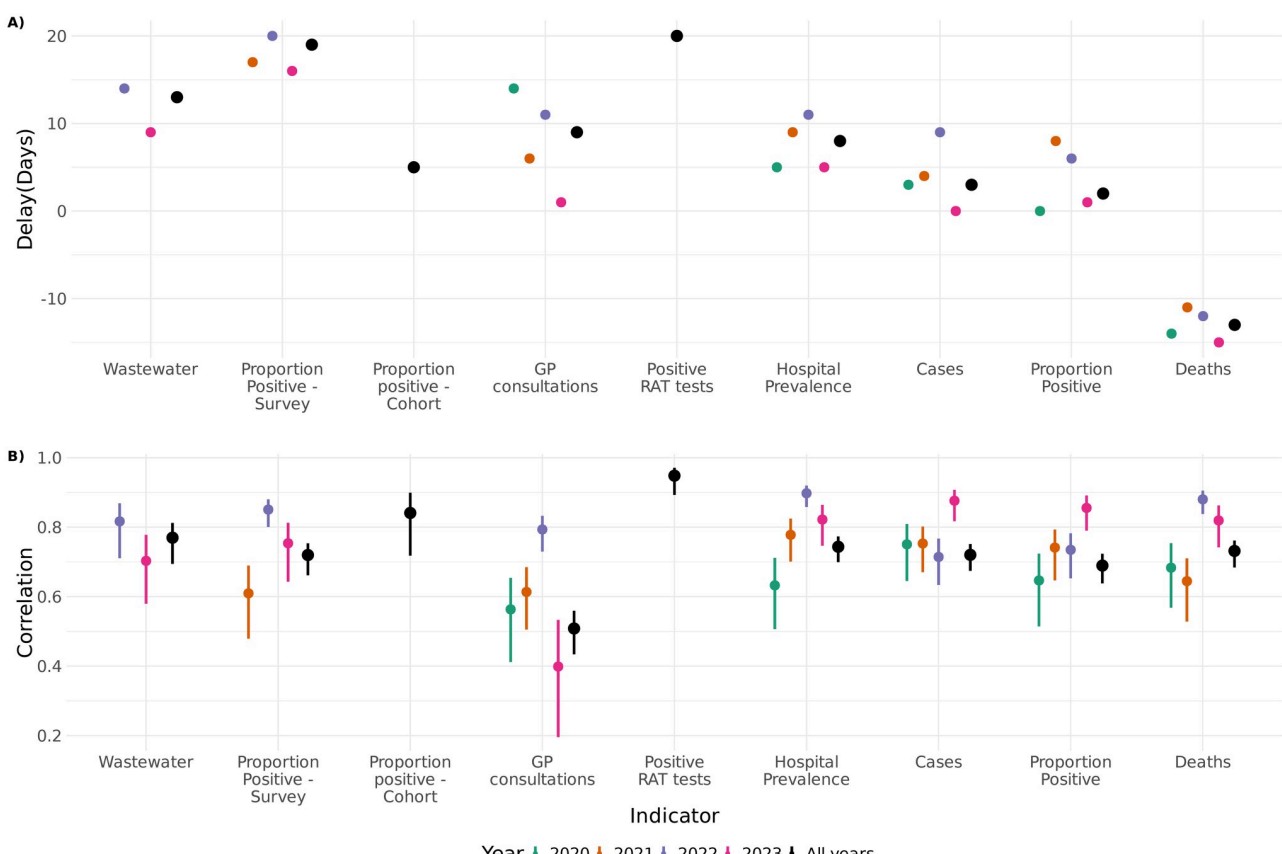

**Fig 2. Estimated time-shifts for each indicator that maximises the correlation with hospitalisations admissions (A) per year and for all years combined and (B) estimated correlation between the indicator and hospital admissions (B).** Positive time-shifts means that the indicator grows prior to hospitalisations and negative time-shift means that they are delayed, Norway 2020–2023.

In Fig 3 we show the estimated and shifted growth rates for each year from 2020 to the end of 2023. These growth rates show the large initial surge in infections in early 2020 followed by a drop after the lockdown in the middle of March. In early 2021, we see high growth rates associated with the Alpha-wave in Norway. The autumn of 2021 sees a resurgence of transmission due to the Delta-variant and later the Omicron(BA1/BA2) wave starting in December 2021. After the large Omicron wave there is another wave in the summer of 2022, dominated by the Omicron BA4/5 variants. In both 2022 and 2023 we see a winter wave with a peak in December followed by a rapid decline.

Apart from early 2020, the growth rate mostly stayed between -10% to 10% indicating that Norway never again experienced such rapid growth as in the early phase of the pandemic. Overall the different indicators give a fairly consistent view of the growth rate. This is also supported by the high correlation coefficients in Fig 4, with no correlations below 0.5 and most being significantly higher. In January 2022, while the growth rate for hospitalisations remained negative, many of the other indicators indicated an escalating pandemic. Additionally, the growth rate for newly confirmed cases becomes negative at least a month before the other indicators.

In Fig 5, we show the combined overall growth rate, the estimated heterogeneity ($I^2$) of the indicators, the estimated reproduction number and estimates of relative incidence based on

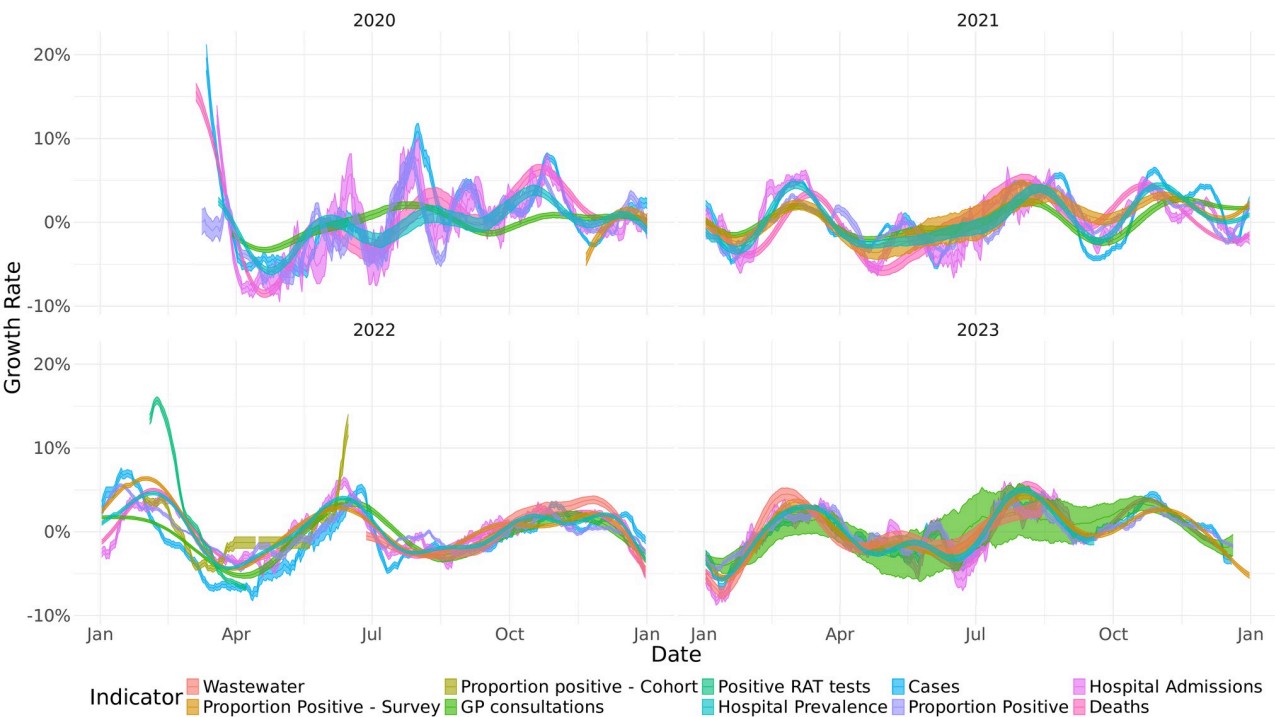

**Fig 3. Estimated growth rates for all surveillance indicators after applying a yearly time-shift(delay) to maximise the correlation with hospital admissions, Norway 2020–2023.** The shaded regions show the 95% confidence intervals.

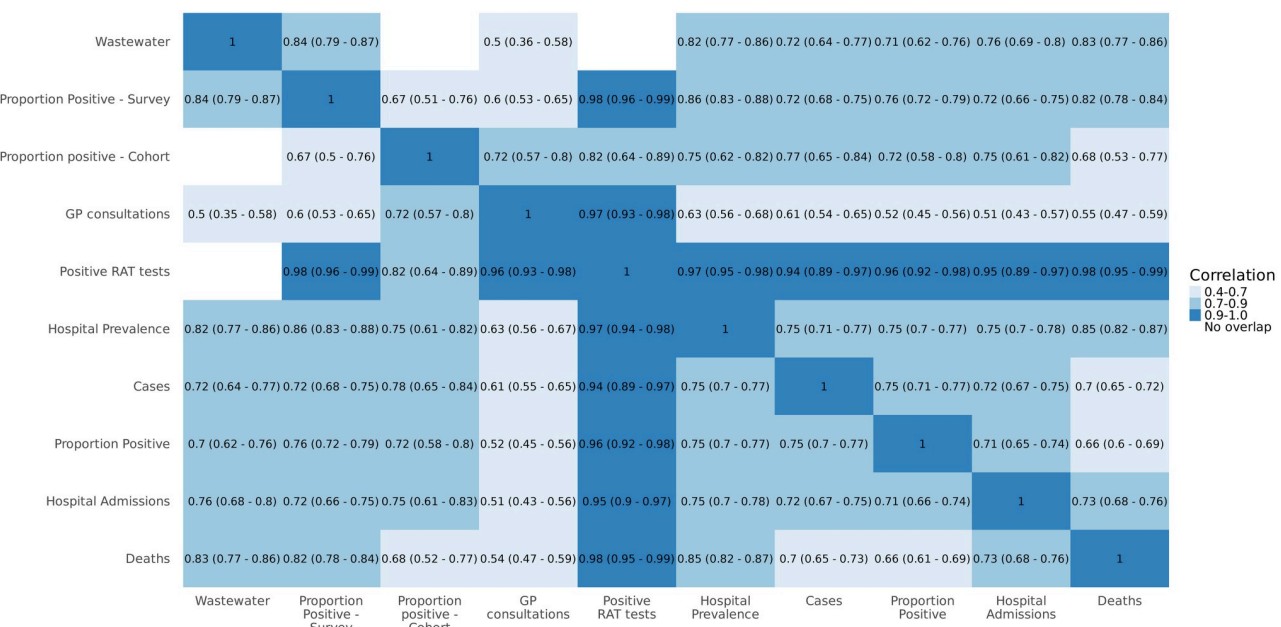

**Fig 4. Estimated Pearson correlation coefficients with 95% confidence intervals between the estimated, time-shifted growth rates of the different surveillance indicators.** Correlations between indicators with no temporal overlap are not calculated and are marked in white, Norway 2020–2023.

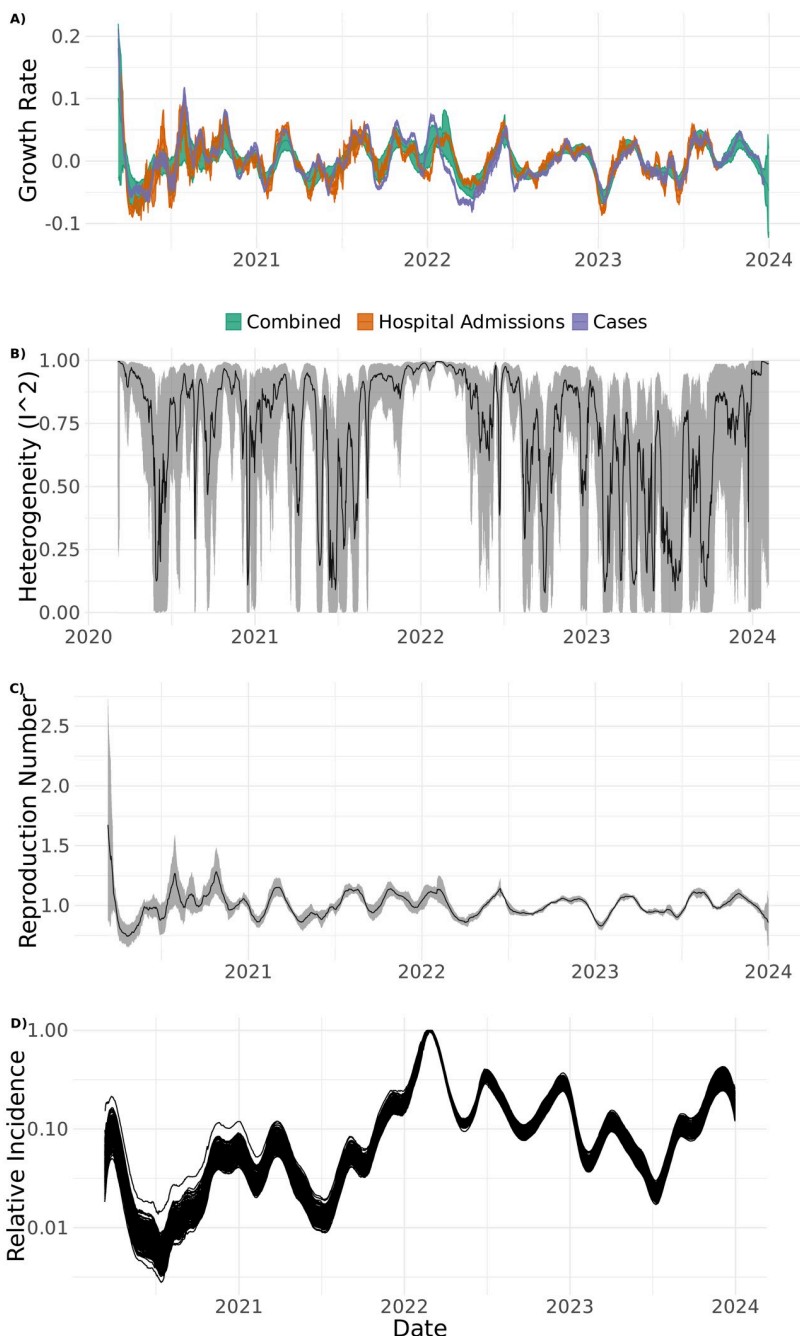

**Fig 5.** A) Estimate of the combined growth rate together with growth rates from hospitalisations and positive cases with 95% confidence intervals; B) estimated heterogeneity between the growth rates from the different indicators; C) estimated reproduction numbers based on the combined growth rate with a 95% confidence intervals; D) shows the estimated relative incidence based on the combined growth rate on a logarithmic scale, Norway 2020–2023.

the growth rates. When comparing the overall growth rate to the rates for cases and hospitalisations we can see multiple times of disagreement, especially around early 2022. The heterogeneity between the growth rates from the different indicators varies significantly over the time with both early 2020 and early 2022 being examples of periods with high heterogeneity. Fig 5D

clearly shows that the Omicron peak in early 2022 was the highest with an incidence more than 10 times greater than observed during 2020.

## Discussion

In this study we demonstrate that estimating and visualising growth rates from many different surveillance indicators can provide a more complete picture of the spread of COVID-19 than only considering laboratory confirmed cases or hospitalisations. We also show that including new indicators from cohort studies, syndromic surveillance, hospital prevalence and wastewater surveillance improves our understanding of the COVID-19 epidemic. This combined approach was published in weekly surveillance reports and provided useful real-time information to inform the pandemic response in Norway [3]. Using multiple surveillance indicators allowed us to obtain more robust trend estimates even when individual indicators were biased due to changes in epidemiology, data collection or policy. This is most clearly evident during the Omicron wave in early 2022 where both the trend estimated from hospitalisations and confirmed cases gives a biased estimate of the growth rate due to changes in severity [6] and changes in testing regimes. During this period, the combined growth rate shows an increasing trend approximately three weeks before the trend in hospitalisations began increasing and the trend in confirmed cases peaked approximately four weeks earlier than the combined growth rate.

The estimated time-shifts that maximised correlations show that all the indicators apart from COVID-19 associated deaths provide an earlier measure of growth than hospitalisation. Assuming similar delays in reporting for different indicators, this is not unexpected, as more severe outcomes usually take some time to develop. This indicates that these data sources are useful for predicting hospitalisations, but for real-time predictions one would need to also take reporting delays and other practical considerations into account. The delays also refer to the estimated incidence and not the directly measured prevalence. The general pattern and size of the delays agree quite well with findings from the UK [18].

We find strong correlations among the estimated growth rates from the different surveillance indicators indicating that they describe the same underlying disease transmission process. However, the weaker correlations between some indicators suggest that they likely measure infections in populations that differ, such as by geographic location or by age. The correlation patterns provide insight into how the different indicators relate to each other and how they can all contribute to a holistic surveillance system. However, they also underscore the importance of repeated point-prevalences studies [16, 17] which would give the best possible estimate of infections in different age groups.

In addition to visualising all the individual growth rates together, we propose a method for combining the individual trends into one overall growth rate with minimal assumptions. By using a random effects meta-analysis framework we can combine all the estimated growth rates and estimate heterogeneity. When the heterogeneity is large, it is crucial to also consider all the individual indicators when interpreting the combined estimate. This combined rate can be translated to a reproduction number. While the early estimates of the reproduction number are very uncertain and difficult to interpret due to large heterogeneity we find that the patterns in the reproduction number estimates agree quite well with other estimates from Norway [4, 5] during the first year of the pandemic. Our estimates are in general somewhat closer to one likely due to a shorter generation time.

In this paper we have used a retrospective approach to estimate trends, utilising complete data that became available after the period for which the trend was initially estimated. This gives a better retrospective estimate of the trend, but is not feasible in real-time without a

long delay. For real-time analysis of trend further research is needed to understand which smoothing techniques provide the best estimate of current trends based on all available data.

The main strengths of the presented approach is that it allows us to translate the different indicators into a common numerical estimate of growth rates that can be visualised together and combined. This allows an holistic description of the trend where it is possible to overcome biases in individual indicators. While the combined rates might be more robust, they still crucially depend on all the individual indicators and their uncertainties and biases will feed into the overall estimates. These biases include changes in testing requirements or health seeking behaviour, changes in reporting and coding practises, changes in severity of disease and non-representative samples. In addition, not all indicators in this study are representative of the general population, making comparisons more difficult. For example, the wastewater data only covers a geographical subset of the population, and the cohort data had a different age profile than the whole country. These differences highlight the benefits of not only presenting one combined estimate, but showing all the estimates from the different sources so that these potential differences can be interpreted. Since our aim is to describe the underlying disease transmission, and not compare or rank indicators based on historical performance or known importance, individual biases are only a problem as far as they impact the overall trend estimate. The proposed methods could also be implemented on a local scale when local data is available.

The simple smoothing method used in this paper will not provide a perfect fit to the underlying data especially for short time resolutions. More advanced smoothing methods based for example on Gaussian Processes [9] could improve the smoothing. The amount of smoothing of the trends corresponding to the time-window used in the regression models in this paper is a key consideration and must balance a trade-off between resolution and statistical uncertainty and the inherent smoothing of the different indicators.

The core of the method is straightforward and could easily and rapidly be implemented in a crisis-situation and it does not depend on any parameters or mechanistic understanding of the disease. It would also be possible to use prevalence data directly instead of the pre-processing step of estimating incidence in the early phases of a crisis. The method can easily be extended to indicators based on other types of data sources including self-reporting of symptoms, web-traffic or mobility data.

In conclusion, using a large number of different surveillance indicators to estimate growth rates that can be interpreted together was an essential part of the surveillance of the COVID-19 pandemic in Norway and can be a powerful tool for routine surveillance and a potential next pandemic.

## Supporting information

**S1 File. Supplementary materials containing additional methodological details and sensitivity analysis.**
(PDF)

## Author Contributions

**Conceptualization:** Gunnar Rø, Karin Nygård, Birgitte Freisleben de Blasio.

**Data curation:** Trude Marie Lyngstad, Elina Seppälä, Siri Nærland Skodvin, Lill Trogstad, Richard Aubrey White, Arve Paulsen, Trine Hessevik Paulsen, Trine Skogset Ofitserova, Petter Langlete, Elisabeth Henie Madslien.

**Formal analysis:** Gunnar Rø.

**Methodology:** Gunnar Rø.

**Software:** Gunnar Rø.

**Visualization:** Gunnar Rø.

**Writing – original draft:** Gunnar Rø.

**Writing – review & editing:** Gunnar Rø, Trude Marie Lyngstad, Elina Seppälä, Siri Nærland Skodvin, Lill Trogstad, Richard Aubrey White, Arve Paulsen, Trine Hessevik Paulsen, Trine Skogset Ofitserova, Petter Langlete, Elisabeth Henie Madslien, Karin Nygård, Birgitte Freisleben de Blasio.

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
