## [Decision Letter · Decision Letter 0]

29 Nov 2024

PONE-D-24-45734Estimating the trend of COVID-19 in Norway by combining multiple surveillance indicatorsPLOS ONE

Dear Dr. Rø,

Thank you for submitting your manuscript to PLOS ONE. After careful consideration, we feel that it has merit but does not fully meet PLOS ONE’s publication criteria as it currently stands. Therefore, we invite you to submit a revised version of the manuscript that addresses the points raised during the review process.

**Your manuscript was reviewed by two experts in the field. Both identified several problems in your submission which require your attention. Please provide point-by-point responses to the attached comments.**

We look forward to receiving your revised manuscript.

Kind regards,

Yury E Khudyakov, PhD

Academic Editor

PLOS ONE

**Journal Requirements:**

2. Please note that your Data Availability Statement is currently missing the repository name. If your manuscript is accepted for publication, you will be asked to provide these details on a very short timeline. We therefore suggest that you provide this information now, though we will not hold up the peer review process if you are unable.

Reviewers' comments:

Reviewer's Responses to Questions

**Comments to the Author**

1. Is the manuscript technically sound, and do the data support the conclusions?

Reviewer #1: Yes

Reviewer #2: Yes

2. Has the statistical analysis been performed appropriately and rigorously? 

Reviewer #1: Yes

Reviewer #2: Yes

3. Have the authors made all data underlying the findings in their manuscript fully available?

Reviewer #1: Yes

Reviewer #2: Yes

4. Is the manuscript presented in an intelligible fashion and written in standard English?

Reviewer #1: Yes

Reviewer #2: Yes

5. Review Comments to the Author

**Reviewer #1:** This paper is methodologically interesting, but I have some concerns about these points:

- The authors characterize the Proportion Positive as an Incidence Indicator, but the definition they provide seems more like a Prevalence indicator, except they refer only to Positive tests in new cases. The definition in the paper states only positive tests divided by total tests, but as it has been seen later, one test can remain positive through different periods of time and ithe authors do not specify how they treat repeated tests.

-Relating to Figure 4 trhe authors should make it clear why some correlations are not included (seemingly because there are no data for those indicators in the same period)

- In the Results section the authors write: "When comparing the overall growth rate to the rates for 226

cases and hospitalisations we can see multiple times of disagreement, especially around 227

early 2022. The heterogeneity between the growth rates from the different indicators 228

varies significantly over the time with both early 2020 and early 2022 being examples of 229

periods with high heterogeneity". Nevertheless they conclude "that estimating and visualising growth rates from many 234

different surveillance indicators can provide a more complete picture of the spread of 235

COVID-19 than only considering laboratory confirmed cases or hospitalisations". As a public health specialist I would prefer to know which of the differente indicators should be included in a robust model for estimating growth rates, as it seems that some of the indiocators are not suitable for a robust model.

**Reviewer #2:** Really I found a problem to understand the title of the article. It doesn't reflects the aim exactly. It should be more obvious. Also the estimate methods are not equivalent in time factor and area distribution which may cause some bias in comparison of data. Also you didn't mention other causes that could affect the growth rate at the same time. Although you said that the ethical approval is not needed. But I think there should be.

6. PLOS authors have the option to publish the peer review history of their article (what does this mean?). If published, this will include your full peer review and any attached files.

Reviewer #1: No

Reviewer #2: **Yes: **Dr Michael Nazmy Agban. Professor of Microbiology and Immunology faculty of medicine assiut university. And head of department of molecular biology institute of Assiut

---

## [Author Response · Author response to Decision Letter 0]

19 Dec 2024

Response to the reviewers

We thank the reviewers for insightful comments and provide a point-by-point answer below.

Reviewer 1

This paper is methodologically interesting, but I have some concerns about these points:

Reviewer Point P 1.1 — The authors characterize the Proportion Positive as an Incidence

Indicator, but the definition they provide seems more like a Prevalence indicator, except they refer

only to Positive tests in new cases. The definition in the paper states only positive tests divided

by total tests, but as it has been seen later, one test can remain positive through different periods

of time and ithe authors do not specify how they treat repeated tests.

Reply: The proportion positive PCR tests was classified as an incidence indicator as it is directly

related to the number of new confirmed PCR cases by dividing by the total number of ”test-events”

per day. Since it is just a rescaling of new cases which is an incidence indicator, we consider proportion

positive to also be an incidence indicator. Both the numerator, number of positive cases, and the

test-event denominator are defined to remove the effect of the same person being tested multiple times.

As described in line 93 (in the manuscript with track changes), test-events only consider one test per

person in a 7 day period and for the number of positive tests we only consider a persons first positive

test within a 60-day period, so repeated positive tests within this period are not included. We have

updated the text in line 89-91 and a minor update in the ”Proportion Positive” row in the table to

clarify this aspect of positive tests.

Reviewer Point P 1.2 — Relating to Figure 4 trhe authors should make it clear why some

correlations are not included (seemingly because there are no data for those indicators in the same

period)

Reply: This is a good point, we have updated the caption for figure 4.

Reviewer Point P 1.3 — In the Results section the authors write: ”When comparing the overall

growth rate to the rates for 226 cases and hospitalisations we can see multiple times of disagreement, especially around 227 early 2022. The heterogeneity between the growth rates from the

different indicators 228 varies significantly over the time with both early 2020 and early 2022 being

examples of 229 periods with high heterogeneity”. Nevertheless they conclude ”that estimating

and visualising growth rates from many 234 different surveillance indicators can provide a more

complete picture of the spread of 235 COVID-19 than only considering laboratory confirmed cases

or hospitalisations”. As a public health specialist I would prefer to know which of the different

indicators should be included in a robust model for estimating growth rates, as it seems that some

of the indicators are not suitable for a robust model.

Reply: This is a good questions and while it would indeed have been useful to be able to determine

which indicators are robust, our aim is instead to try to use all the available surveillance indicators to

understand the underlying transmission process. Even less robust surveillance indicators can provide important information. During the pandemic all indicators had a changing relationship with the underlying

infections, and that the best solution therefore is to consider all available indicators holistically as we

show in figure 3 with all the growth rates. Using the meta-analysis model also allows us to combine the

different growth rates into one when the heterogeneity is low, and gives us a signal that interpretation is

more difficult when heterogeneity is high. In these situations, one needs real epidemiological and public

health insight into the disease processes, and the data collection and interpretation of each individual

indicator to determine which indicator provides the best description of the underlying transmission. We

have tried to clarify our aims in line 58-73, a minor edit in the abstract in line 5 and in the discussion

in line 312-315.

Reviewer 2

Reviewer Point P 2.1 — Really I found a problem to understand the title of the article. It

doesn’t reflects the aim exactly. It should be more obvious.

Reply: We apologies if the aim was not described well enough. The aim is to use multiple surveillance

indicators to provide a more comprehensive and robust estimate the underlying trend of new COVID19 infections in Norway. We have tried to highlight this in line 58-73 (In the manuscript with track

changes) and with a minor edit in the abstract in line 5. We hope that this makes it clear that the title

corresponds to our aims in the article.

Reviewer Point P 2.2 — Also the estimate methods are not equivalent in time factor and area

distribution which may cause some bias in comparison of data. Also you didn’t mention other

causes that could affect the growth rate at the same time.

Reply: We completely agree that there are differences in geographical area, age of participants and

many other characteristics between the different surveillance indicators. This is discussed in line 302-

310. As we commented on above, the goal of the paper is not to compare and contrast the different

indicators, but to extract useful estimates of the overall trend in new infections. The high correlation

between the indicators clearly show that even if there are differences between the different indicators,

they also all provide information about the trend in the underlying transmission process. We have

clarified this further in the lines 312-315.

Reviewer Point P 2.3 — Although you said that the ethical approval is not needed. But I

think there should be.

Reply: The data we have used contain no identifiable data and only consist of the aggregated number of

events (cases/admission/deaths/etc) on a national level. The data has been collected and prepared for

surveillance purposes and have already been published in weekly reports by the Norwegian Public Health

Institute(https://www.fhi.no/publ/statusrapporter/luftveisinfeksjoner/) and in part in many other publications(i.e https://github.com/folkehelseinstituttet/surveillance data). Since this is just a reanalysis of

data already collected, used and published for public health surveillance purposes, no additional ethical

approval is needed. We have clarified this in line 208.

---

## [Editor Report · Decision Letter 1]

22 Dec 2024

Estimating the trend of COVID-19 in Norway by combining multiple surveillance indicators

PONE-D-24-45734R1

Dear Dr. Rø,

We’re pleased to inform you that your manuscript has been judged scientifically suitable for publication and will be formally accepted for publication once it meets all outstanding technical requirements.

Kind regards,

Yury E Khudyakov, PhD

Academic Editor

PLOS ONE
---

## [Editor Report · Acceptance letter]

20 Jan 2025

PONE-D-24-45734R1 

PLOS ONE

Dear Dr. Rø, 

I'm pleased to inform you that your manuscript has been deemed suitable for publication in PLOS ONE. Congratulations! Your manuscript is now being handed over to our production team.

Kind regards, 

on behalf of

Dr. Yury E Khudyakov 

Academic Editor

PLOS ONE